

# Effects of vegetation restoration on soil quality in fragile karst ecosystems of southwest China

Huiling Guan[1,2] and Jiangwen Fan[1]

[1] Key Laboratory of Land Surface Pattern and Simulation, Institute of Geographic Sciences and Natural Resources Research, Chinese Academy of Sciences, Beijing, Beijing, China
[2] University of Chinese Academy of Sciences, Beijing, Beijing, China

## ABSTRACT

Soil quality assessment is important for karst ecosystems where soil erosion is significant. A large amount of vegetation restoration has been implemented since the early 21st century in degraded karst areas across southwestern China. However, the impacts on soil quality of different restoration types rarely have been compared systematically. In the current study, we investigated the soil quality after a number of vegetation restoration projects as well as their adjacent cropland by analyzing soil samples. Six vegetation restoration types were evaluated, including one natural restoration (natural shrubland, protected for 13 years), three economic forests (4 years *Eucalyptus robusta*, 4 years *Prunus salicina* and 6 years *Zenia insignis*) and two mixed forests (1 year *Juglans regia*–crop and 13 years *Toona sinensis-Pennisetum purpureum* ). We evaluated the benefits of different restoration types more accurately by setting each adjacent cropland as the control and setting the variation between the corresponding restored and control site as the evaluation object so that the background differences of six sites could be eliminated. The results indicated that natural shrubland, *Toona sinensis-Pennisetum purpureum* and *Zenia insignis* were effective in improving soil quality index (SQI) in degraded karst cropland largely due to their higher SOC and TN content. The variation of SQI (VSQI) of natural shrubland was significantly higher than that in *Eucalyptus robusta*, *Prunus salicina* and *Juglans regia*-crop in total soil layer (0–30 cm) ($P < 0.05$), indicating natural shrubland had better capacity to improve soil quality. The boosting regression tree model revealed that vegetation restoration type explained 73.49% and restoration time explained 10.30% of the variation in VSQI, which confirmed that vegetation restoration type and restoration time are critical for achieving soil reserves. Therefore, it is vital to select appropriate vegetation type in restoration projects and recovery for a long time in order to achieve better soil quality. The current study provides a theoretical basis on which to assess the effects of different vegetation restoration types on the heterogeneous degraded karst areas.

Corresponding author
Jiangwen Fan, fanjw@igsnrr.ac.cn

## INTRODUCTION

Southwestern China has the largest continuous karst landscape in the world, spanning an area of about 510,000 km$^2$ (*Li et al., 2018c*). This region is characterized by shallow soil due mainly to slow soil formation rate from limestone (*Peng & Wang, 2012*; *Zhao et al., 2017*). In addition, the steep and broken surface, the seasonal and abundant precipitation, and decades of poorly managed intensive agriculture occurring in this area all contribute toward exacerbating soil loss (*Cheng et al., 2017*; *Peng et al., 2018*; *Yan et al., 2018*).

To address these problems, several national-scale ecological restoration projects have been carried out in karst areas, including the Grain for Green Program, the Rocky Desertification Control Project and the Natural Forest Protection Project (*Zhang et al., 2016*, *2018b*). The recovery of soil functions is vital to ecosystem regeneration of degraded croplands (*Guo et al., 2019*), many scholars have evaluated the impacts of different vegetation restoration strategies on soil quality, which has been widely used to determine how soil responds to various management practices (*Raiesi & Kabiri, 2016*; *Guo et al., 2018*; *Vincent et al., 2018*). However, most of these studies compared the soil quality of different vegetation restoration types without considering their original ecosystem conditions, and judged each type based only on the status quo (*Yang et al., 2017*; *Li et al., 2018a*; *Zhang et al., 2019a*). In fact, vegetation restoration types were taken according to different karst environments. For example, artificial grassland, shrub and forest correspond to moderate-, light- and non-karst rocky desertification, respectively (*Li et al., 2009*); that is to say, the environmental backgrounds of different vegetation restoration types are different. Thus, the soil quality of each vegetation restoration type cannot truly reflect its effects because factors such as topography, the degree of rocky desertification (*Sheng et al., 2018*) and basic soil formation factors (*Karlen, Ditzler & Andrews, 2003*) also affect the soil quality of each site. Hence, there is an urgent need to compare the effects of different vegetation restoration types properly, using corresponding unrestored control sites for comparison.

In this present study, we measured soil quality parameters associated with different vegetation restoration types as well as those of corresponding adjacent unrestored croplands, and determined the difference value between the soil quality of the paired treated and untreated sites as the evaluation object. Since each paired adjacent restored and unrestored site has consistent soil parent material, climate and topographic conditions, our evaluation object could eliminate the effects of those environmental factors and only reflect the effects of vegetation restoration. Therefore, our results will be able to assess the difference in effects of different vegetation types that distributed in different sites on soil quality more accurately.

We established soil quality index (SQI) based on the Minimum Data Set (MDS) approach, which is an effective and dependable method of assessing soil quality (*Lin et al., 2017*; *Nabiollahi et al., 2018*). The objectives of the current study were (1) to assess the variation in soil properties caused by different vegetation restoration types, (2) to calculate SQI and evaluate the effects of vegetation restoration compared to cropland, (3) to calculate the variation in SQI between restoration and its respective control (VSQI), in

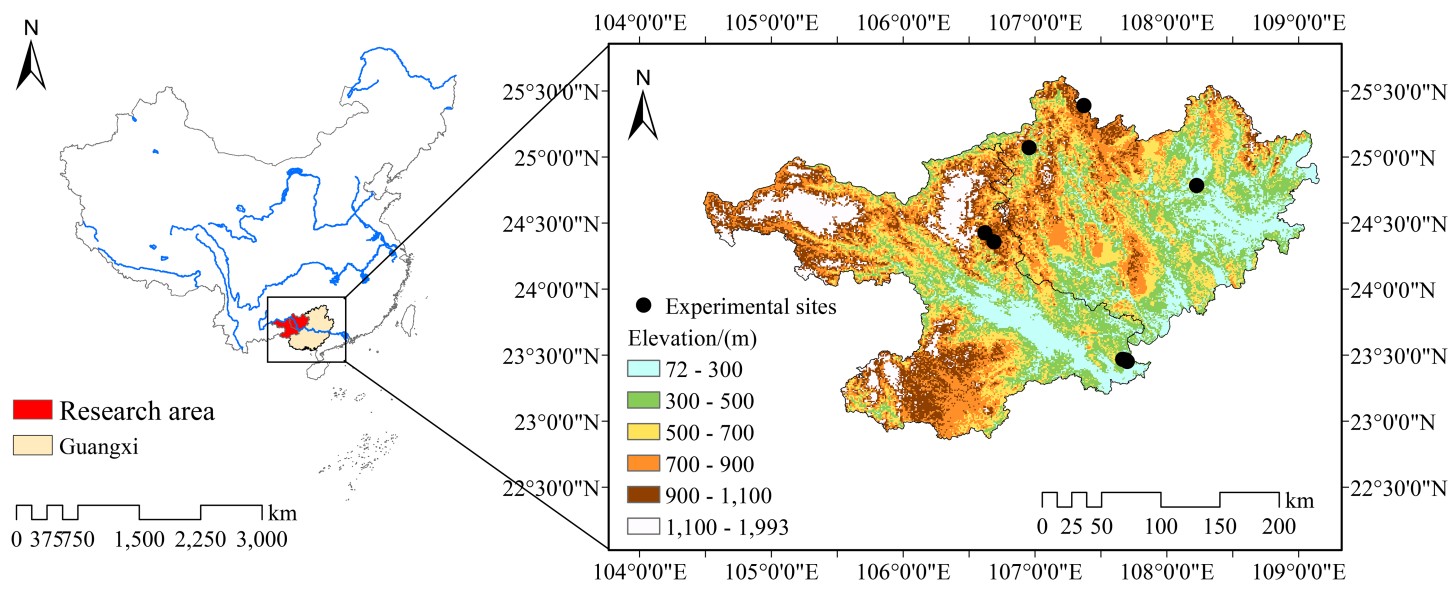

**Figure 1** **Location of study sites.** From the easternmost counterclockwise point are *Eucalyptus robusta* economic forest, *Juglans regia*–crop mixed forest, *Prunus salicina* economic forest, *Toona sinensis–Pennisetum purpureum* mixed forest, natural shrubland and *Zenia insignis* economic forest, respectively. Data: Resource and Environment Data Cloud Platform (http://www.resdc.cn/).

order to compare the effects of different vegetation types on soil quality using the VSQI value, and (4) to identify the factors that influenced VSQI.

# MATERIALS AND METHODS

## Study area

This study was conducted in two eroded hilly karst cities, Hechi (23°41′–25°37′N; 106°34′–109°09′) and Baise (22°51′–25°07′N; 104°28′–107°54′E), located in the northwest of Guangxi Province, Southwest China (Fig. 1). Before 2003, the sampling sites were sloped cropland, which had a serious problem of rocky desertification. From 2003, the Chinese government and researchers gradually began to perform various vegetation restoration projects in this area. This area is characterized by a subtropical monsoon climate, and it has an average annual temperature of 18.39 °C and an average annual rainfall of 1,347.88 mm. It can be divided into a rainy season (April–August) and a dry season (October–March) each year (*Li et al., 2017*). Topography is high in the Northwest and Southwest, but low in the South and East. The soil is dominated by calcareous lithosols over both limestone and dolomite and their mixtures (*Xiao et al., 2018*).

## Soil sampling and laboratory analyses

In April 2018, six vegetation restoration types were selected for study: including one natural restoration (natural shrubland, protected for 13 years), three economic forests (4 years *Eucalyptus robusta*, 4 years *Prunus salicina* and 6 years *Zenia insignis*) and two mixed forests (1 year *Juglans regia*-crop and 13 years *Toona sinensis-Pennisetum purpureum* (napier grass)), they are natural shrubland, *Eucalyptus robusta*, *Prunus salicina*, *Zenia insignis*, *Juglans regia*-crop and *Toona sinensis-Pennisetum purpureum* for

**Table 1 Basic profiles of experimental plots.**

| Vegetation restoration types | Natural shrubland | *Eucalyptus robusta* | *Prunus salicina* | *Zenia insignis* | *Juglans regia-crop* | *Toona sinensis-Pennisetum purpureum* |
|---|---|---|---|---|---|---|
| Study sites | Baise | Hechi | Hechi | Baise | Hechi | Baise |
| Area/ha | 13 | 20 | 26.7 | 6.7 | 20 | 13.3 |
| Elevation/m | 653 | 243 | 488 | 229 | 846 | 740 |
| Slope/° | 15 | 20 | 15 | 30 | 20 | 25 |
| Mean tree height/m | 2.1 | 10 | 2.5 | 8 | 2.3 | 12 |
| Vegetation cover/% | 70 | 60 | 50 | 70 | 20 | 20 |
| Before restoration | cropland | cropland | cropland | cropland | cropland | cropland |
| Recovery time/year | 13 | 4 | 4 | 6 | 1 | 13 |
| Main species | Cyclobalanopsis glauca, Sapium sebiferum, Choerospondias axillaris, Mallotus japonicas, Cryptocarya chinensis, Eriobotrya japonica, Cinnamomum japonicum, Mahonia fortune, Smilax china, Rubus corchorifolius, Nephrolepis auriculata | Eucalyptus robusta, Cayratia japonica, Bidens pilosa, Youngia japonica, Conyza canadensis, Dioscorea opposita, Mallotus japonicus, Clinopodium chinensis, Carpesium abrotanoides, Loropetalum chinensis, Trachelospermum jasminoides, Ophiopogon japonicus, Viola verecunda, Selaginella uncinata | Prunus salicina, Conyza Canadensis, Oxalis corniculata, Bidens pilosa, Lespedeza bicolor, Argyreia seguinii, Amaranthus tricolor, Imperata cylindrica, Sonchus oleraceus, Centaurea cyanus, Rubus corchorifolius | Zenia insignis, Bidens pilosa, Arthraxon hispidus, Mallotus japonicus, Dendranthema indicum | Juglans regia, Zea mays, Glycine max, Arthraxon hispidus, Oxalis corniculata, Hydrocotyle sibthorpioides, Kalimeris indica, Silybum marianum | Toona sinensis, Pennisetum purpureum, Zea mays, Rubus corchorifolius, Conyza Canadensis, Youngia japonica, Artemisia argyi, Galium aparine, Mallotus japonicas, Eupatorium adenophora, Viola verecunda, Bidens pilosa, Commelina communis, Nephrolepis auriculata, Gnaphalium affine, Oxalis pescaprae, Rubus corchorifolius |

short, respectively. Three sites (*Eucalyptus robusta*, *Prunus salicina*, *Juglans regia*-crop) were set up in Hechi city, while the other three sites were in Baise city (Table 1). In order to ensure comparability, three 4 × 4 m plots were selected randomly and each plot is spaced more than 10 m apart, for assessment of each vegetation type.

Meanwhile, cropland site adjacent to each vegetation restoration site was taken as a paired control. In total, 36 plots were chosen for field observation and sampling. Before sampling, we were approved and supported by farmers and nature reserve managers of
these sites. In every plot, the litter layer was removed and soil samples were collected in three replicate sub-plots from three layers (0–10, 10–20 and 20–30 cm). Samples in the same soil layer from the same subplots were mixed and sieved (<2 mm), removing roots and stone. A subsample of fresh soil was stored at −20 °C for subsequent available nitrogen analyses, other subsamples were air dried, with portions sieved to 0.147 mm.

In the laboratory, the following soil physical and chemical properties were measured according to *Pang et al. (2018)* and *Chen et al. (2013)*: soil pH, soil organic carbon (SOC), total carbon (TC), total nitrogen (TN), total phosphorus (TP), total potassium (TK), ammonium–nitrogen ($NH_4^+$), nitrate–nitrogen ($NO_3^-$), available potassium (AK), and available phosphorus (AP).

## Evaluation of the effects of vegetation restoration on soil quality index

SQI is widely used to evaluate soil quality. It consists of four procedures: (1) selection of minimum data set (MDS); (2) standardization of each MDS parameter; (3) weighting of each MDS parameter based on a principal component analysis (PCA); and (4) calculation of SQI by merging the scores (*Zhang et al., 2011*).

Two steps ensure that MDS indicators are more representative and exhibit less redundancy. Firstly, PCA was performed (*Doran & Parkin, 1994*). We took into consideration for the MDS only those principal components (PCs) with eigenvalues ≥ 1 (*Ye, Cheng & Zhang, 2014*) and which explained more than 5% of the total variation (*Andrews et al., 2003*). For each PC, indexes with the maximum weight and over 90% of the maximum weight were selected (*Askari & Holden, 2014*). Then, Pearson's correlation analysis was adopted to check whether other indicators should be removed if there were more than one high loading indicators in a single PC (*Bastida et al., 2006*). Wherever indicators within a PC were highly correlated with one another (correlation coefficient > 0.6), we selected only the indicator with the highest eigenvector (*Andrews, Karlen & Cambardella, 2004*).

To eliminate the differences in indicators units, a nonlinear scoring method was used to score soil indicators to a value between 0 and 1.0. The sigmoidal function (Eq. (1)) was performed as follows (*Zhang et al., 2019a*):

$$S_i = a \Big/ \left[ 1 + (x_i/x_{0i})^b \right] \qquad (1)$$

Where *i* refers to an indicator in MDS, $S_i$ is the score of the *i* soil indicator, *a* is the maximum score (*a* =1), $x_i$ is every measured value of the *i* indicator, $x_{0i}$ is the mean value of the *i* soil indicator, and *b* is the value of the equation's slope; *b* = −2.5 was applied to a 'more is better' curve and *b* = 2.5 was applied to a 'less is better' curve, respectively (*Bastida et al., 2006*).

SQI was calculated as follows (*Zhang et al., 2019a*):

$$SQI = \sum_{i=1}^{n} S_i \times W_i \qquad (2)$$

Where $W_i$ is the weighting values of the MDS determined by PCA, $S_i$ is the indicator score based on Eq. (1), and $n$ is the number of the selected indicators in MDS. We assumed that higher SQI values indicated superior soil functions or better soil quality.

Variation in vegetation restoration types on soil quality index was calculated using the following equation:

$$\text{VSQI}_j = SQI_{pj} - SQI_{ckj} \tag{3}$$

where $\text{VSQI}_j$ is the changed value achieved by undertaking vegetation restoration $j$, $SQI_{pj}$ is the SQI of vegetation restoration $j$, and $SQI_{ckj}$ is the SQI of its control.

### Statistical analysis

All data are presented as the means ± standard error. One-way analyses of variation followed by the Tukey pairwise multiple comparison test was used to assess the differences in the soil physicochemical and SQI values among different vegetation restoration types and different soil layers at the $P < 0.05$ level. Paired sample $t$ test was used to evaluate the differences between restored project and adjacent unrestored cropland. PCA and Pearson's correlation analysis were used to select the soil indicators and to weight the selected indicators. Boosting regression tree model (BRT) was used to reflect the contribution of each factor and was carried out with R (R 3.50) using the gbm.step function from the dismo package in R to determine the variation in SQI that was explained by each indicator. All statistical analyses were performed by IBM SPSS 22 (IBM, Armonk, NY, USA). Figures were generated using Origin 2018b (Origin Lab., Hampton, MA, USA).

## RESULTS

### Effects of vegetation restoration on soil physicochemical properties

Paired sample t test evaluated the significance of vegetation restoration to soil properties, by directly comparing the result from each treated replicate plot with the corresponding adjacent untreated replicate plot (Tables 2 and 3). Natural shrubland, *Prunus salicina* and *Toona sinensis-Pennisetum purpureum* had no significant effect on soil texture, while *Eucalyptus robusta*, *Zenia insignis* and *Juglans regia*-crop could significantly increase sand content in 0–10 and 10–20 cm soil layer, accordingly, decrease clay content ($P < 0.05$). Vegetation restoration could significantly affect TN content and pH ($P < 0.05$), changing in opposite directions, with the exception of *Toona sinensis-Pennisetum purpureum*. *Eucalyptus robusta* could significantly decrease soil pH, occurred in every soil layer ($P < 0.05$). Although most restoration types increased TN content significantly ($P < 0.05$), except for *Prunus salicina* and *Juglans regia*-crop, the C/N ratio still showed a significant increase under all restoration types ($P < 0.01$), except for *Eucalyptus robusta*. In addition, natural shrubland and *Toona sinensis-Pennisetum purpureum* increased SOC content significantly ($P < 0.05$), but barely increased available nutrients (AP, AK, $NH_4^+$ and $NO_3^-$). On the contrary, *Juglans regia*–crop increased AP and $NO_3^-$ content significantly ($P < 0.05$).

Variation between treated restoration plot and the corresponding adjacent untreated plot can be used to compare the effects of different restoration types on soil physicochemical

**Table 2 Paired *t* test values for effect on soil texture (clay, silt, sand) between vegetation restoration type and the control.**

| Set | Soil texture | 0–30 cm | 0–10 cm | 10–20 cm | 20–30 cm |
|---|---|---|---|---|---|
| NS–CK1 | Clay | | | | |
| | Silt | | | | |
| | Sand | | | | |
| EF–CK2 | Clay | − | − | − | |
| | Silt | + | ++ | + | |
| | Sand | + | | | |
| PF–CK3 | Clay | | | | |
| | Silt | | | | |
| | Sand | | | | |
| ZF–CK4 | Clay | −− | − | − | − |
| | Silt | ++ | + | + | + |
| | Sand | ++ | + | + | |
| JC–CK5 | Clay | −− | −− | − | |
| | Silt | −− | −− | − | − |
| | Sand | ++ | ++ | + | |
| TG–CK6 | Clay | | | | |
| | Silt | | | | |
| | Sand | | | | |

**Note:**

++ or −−: Difference is significant at *P* < 0.01 level in double-tailed *t* test. +or− : Difference is significant at *P* < 0.05 level in double-tailed *t* test. + indicates a significant increase and − indicates a significant decrease, empty shows no significant change. NS, natural shrubland; EF, *Eucalyptus robusta* economic forest; PF, *Prunus salicina* economic forest; ZF, *Zenia insignis* economic forest; JC, *Juglans regia*-crop mixed forest; TG, *Toona sinensis-Pennisetum purpureum* mixed forest. *n* = 3 in every soil layer, *n* = 9 in total soil layer (0–30 cm) for 6 restoration types except for EF, which *n* = 4 in every soil layer and *n* = 12 in total soil layer (0–30 cm). The same below.

properties. Variation in sand content under *Juglans regia*-crop (13.37–21.32%) was significantly higher than that under all other vegetation restoration types at 0–10 and 10–20 cm soil layers (Table 4). Silt content variation under *Juglans regia*-crop (10.0–17.19%) was significantly lower than that for the other types at the same two soil depths, except for natural shrubland. Variation in SOC (VSOC) over the total soil depth (0–30 cm) was in the range −2.56–18.10 g/kg. The VSOC under *Prunus salicina* was significantly lower than that under *Toona sinensis-Pennisetum purpureum* and natural shrubland (*P* < 0.05; Figs. 2A–2D), indicating a better SOC accumulation capacity in the latter 2 restoration types. Similarly, the variation in TN (VTN) of *Prunus salicina* was significantly lower than those of all other types (*P* < 0.05; Figs. 2E–2H), except for *Juglans regia*-crop. In addition, significant differences in VSOC and VTN among soil depths were found only in *Prunus salicina* and *Eucalyptus robusta*, respectively, with the values decreasing with increasing soil depth (*P* < 0.05). However, no significant differences of the variation in C/N ratio (VC/N) (0–30 cm) were observed among different vegetation restoration types. The VC/N of *Prunus salicina* and natural shrubland increased significantly with greater soil depth (*P* < 0.05; Figs. 2I–2L).

**Table 3 Paired *t* test values for effect on soil indicators between vegetation restoration type and the control.**

| Set | Soil layer | SOC | TN | TP | TK | AP | AK | $NH_4^+$ | $NO_3^-$ | C/N | pH |
|---|---|---|---|---|---|---|---|---|---|---|---|
| NS–CK1 | 0–30 | ++ | ++ | ++ | | − | | | −− | ++ | −− |
| | 0–10 | | + | ++ | −− | | | | − | | |
| | 10–20 | | + | | | − | | | − | | − |
| | 20–30 | | + | | | | | | − | | |
| EF–CK2 | 0–30 | | ++ | | | | | + | | | −− |
| | 0–10 | | + | | | | | + | | | −− |
| | 10–20 | | | | | | | + | − | | −− |
| | 20–30 | | + | | | | − | + | | | −− |
| PF–CK3 | 0–30 | | −− | | −− | | + | −− | | ++ | ++ |
| | 0–10 | | | | − | | | | | | |
| | 10–20 | | − | | −− | | | | | | |
| | 20–30 | | − | − | | | | − | | | |
| ZF–CK4 | 0–30 | | ++ | −− | −− | − | | ++ | | ++ | −− |
| | 0–10 | | | − | − | − | | | | | |
| | 10–20 | | + | −− | −− | − | − | + | | | |
| | 20–30 | | ++ | | −− | | | ++ | | | |
| JC–CK5 | 0–30 | | − | −− | −− | + | −− | − | ++ | ++ | ++ |
| | 0–10 | | | − | −− | | | | | | |
| | 10–20 | | | −− | −− | | | | −− | | + |
| | 20–30 | | | − | | ++ | − | | | + | |
| TG–CK6 | 0–30 | ++ | ++ | | −− | | | | −− | ++ | ++ |
| | 0–10 | | + | | − | | | − | | | |
| | 10–20 | | + | | | | | | | | + |
| | 20–30 | | + | | −− | | − | | − | | + |

Note:
SOC, soil organic carbon; TN, total nitrogen; C/N, the ratio of SOC to TN; TP, total phosphorus; TK, total potassium; AP, available phosphorus; AK, available potassium; $NH_4^+$, ammonium-nitrogen; $NO_3^-$, nitrate-nitrogen. ++ or −−: Difference is significant at $P < 0.01$ level in double-tailed *t* test. + or −: Difference is significant at $P < 0.05$ level in double-tailed *t* test. + indicates a significant increase and − indicates a significant decrease, empty shows no significant change.

According to the variation in pH (0–30 cm) (VpH), the vegetation restoration types can be divided into three categories: (1) VpH was positive (0.73–0.90), including *Prunus salicina*, *Juglans regia*-crop and *Toona sinensis-Pennisetum purpureum*; (2) pH was slightly reduced (−0.47 to −0.29), including natural shrubland and *Zenia insignis*; and (3) pH was greatly reduced (−2.53), including only one restoration type (*Eucalyptus robusta*). Significant differences were found among the three categories ($P < 0.05$; Figs. 2M–2P). The significant difference of VpH in response to greater soil depth was found only in *Eucalyptus robusta*, under which VpH decreased with greater soil depth ($P < 0.05$).

Variation in TP (VTP) (0–30 cm) was in the range −1.29–0.18 g/kg, natural shrubland was the only vegetation restoration type with a positive VTP, which was significantly higher than all other five types ($P < 0.05$; Fig. 2T). VTP in *Zenia insignis* was significantly lower than those for the other types at 0–10 cm and 10–20 cm soil depth ($P < 0.05$;

**Table 4 Soil texture (clay, silt, sand) analysis for variation in different vegetation restoration types.**

| Types | Depth | Clay (%) | Silt (%) | Sand (%) |
|---|---|---|---|---|
| NS | 0–10 cm | 2.01 ± 0.56Aa | −1.69 ± 0.58Ab | −0.33 ± 0.26Ab |
|  | 10–20 cm | 2.35 ± 1.03Aa | −2.07 ± 1.61Abc | −0.28 ± 0.68Ab |
|  | 20–30 cm | 1.89 ± 0.36Aa | −1.84 ± 0.41Aab | −0.05 ± 0.20Ab |
| EF | 0–10 cm | −4.63 ± 1.76Ac | 4.53 ± 1.40Aa | 0.10 ± 0.48Ab |
|  | 10–20 cm | −5.72 ± 3.46Ac | 4.28 ± 2.51Aa | 1.45 ± 0.97Ab |
|  | 20–30 cm | 0.83 ± 2.44Aa | −1.38 ± 2.78Bab | 0.55 ± 1.06Ab |
| PF | 0–10 cm | −0.38 ± 3.11Bab | −0.12 ± 1.79Ab | 0.50 ± 1.35Ab |
|  | 10–20 cm | −1.68 ± 2.65Babc | 1.30 ± 2.04Aab | 0.39 ± 0.61Ab |
|  | 20–30 cm | −3.17 ± 1.77Aab | 2.50 ± 1.43Aa | 0.67 ± 0.34Aab |
| ZF | 0–10 cm | −5.85 ± 1.12Ac | 4.70 ± 0.91Aa | 1.15 ± 0.21Ab |
|  | 10–20 cm | −5.89 ± 0.85Ac | 4.81 ± 0.68Aa | 1.08 ± 0.17Ab |
|  | 20–30 cm | −5.17 ± 0.71Ab | 4.02 ± 0.72Aa | 1.15 ± 0.14Aab |
| JC | 0–10 cm | −4.13 ± 0.54Abc | 4.70 ± 2.10Ac | 21.32 ± 2.10Aa |
|  | 10–20 cm | −4.44 ± 2.17Abc | 4.81 ± 1.92Ac | 20.68 ± 4.08Aa |
|  | 20–30 cm | −3.35 ± 2.60Aab | 4.02 ± 4.40Ab | 13.37 ± 5.27Aa |
| TG | 0–10 cm | −4.04 ± 0.98Abc | 1.26 ± 1.81Aab | 2.78 ± 2.25Ab |
|  | 10–20 cm | 0.92 ± 1.91Aab | −0.65 ± 1.50Ab | −0.26 ± 0.42Ab |
|  | 20–30 cm | −1.51 ± 3.16Aab | −4.85 ± 7.88Aab | 6.36 ± 11.03Aab |

Note:
NS ($n$ = 3), natural shrubland; EF ($n$ = 4), *Eucalyptus robusta* economic forest; PF ($n$ = 3), *Prunus salicina* economic forest; ZF ($n$ = 3), *Zenia insignis* economic forest; JC ($n$ = 3), *Juglans regia*-crop mixed forest; TG ($n$ = 3), *Toona sinensis-Pennisetum purpureum* mixed forest. Different lowercase letters indicate significant difference among different vegetation restoration types at the same depth (one–way ANOVA, $P$ < 0.05) and different uppercase letters indicate significant difference under different soil depths at the same type (one–way ANOVA, $P$ < 0.05). The same below.

Figs. 2Q and 2R), indicating that TP decreased greatly under *Zenia insignis*. The absolute value of VTP for *Prunus salicina* and *Zenia insignis* decreasing significantly with greater soil depth ($P$ < 0.05; Figs. 2Q–2S). Variation in TK (VTK) under *Juglans regia*-crop was significantly lower than that under the other restoration types, followed by *Prunus salicina* ($P$ < 0.05; Figs. 2U–2X). Variation in AP (VAP) (0–30 cm) was in the range −15.82–3.52 mg/kg. The VAP values under *Toona sinensis-Pennisetum purpureum*, natural shrubland and *Zenia insignis* were significantly lower than those under *Juglans regia*-crop and *Eucalyptus robusta* ($P$ < 0.05; Fig. 2BB). The absolute value of VAP decreased with greater soil depth, with significant differences among the various soil depths being observed under natural shrubland, *Zenia insignis* and *Prunus salicina* ($P$ < 0.05; Figs. 2Y, 2Z and 2AA). No significant differences were observed among restoration types at the deepest soil layer (20–30 cm). *Prunus salicina* was the only restoration type with positive variation in AK (VAK) (0–30 cm), being significantly higher than that under *Zenia insignis*, *Juglans regia*-crop and *Toona sinensis-Pennisetum purpureum* ($P$ < 0.05; Fig. 2FF). Similar to VAP, the absolute value of VAK had a tendency to decrease with soil depth, with the exception of *Eucalyptus robusta* (Figs. 2CC–2EE) and the absolute value of variation in $NO_3^-$ ($VNO_3^-$) (0–30 cm) decreased significantly with soil depth, except for *Zenia insignis* and *Juglans regia*-crop ($P$ < 0.05; Figs. 2GG–2II). Variation in $NH_4^+$ ($VNH_4^+$) (0–30 cm) was

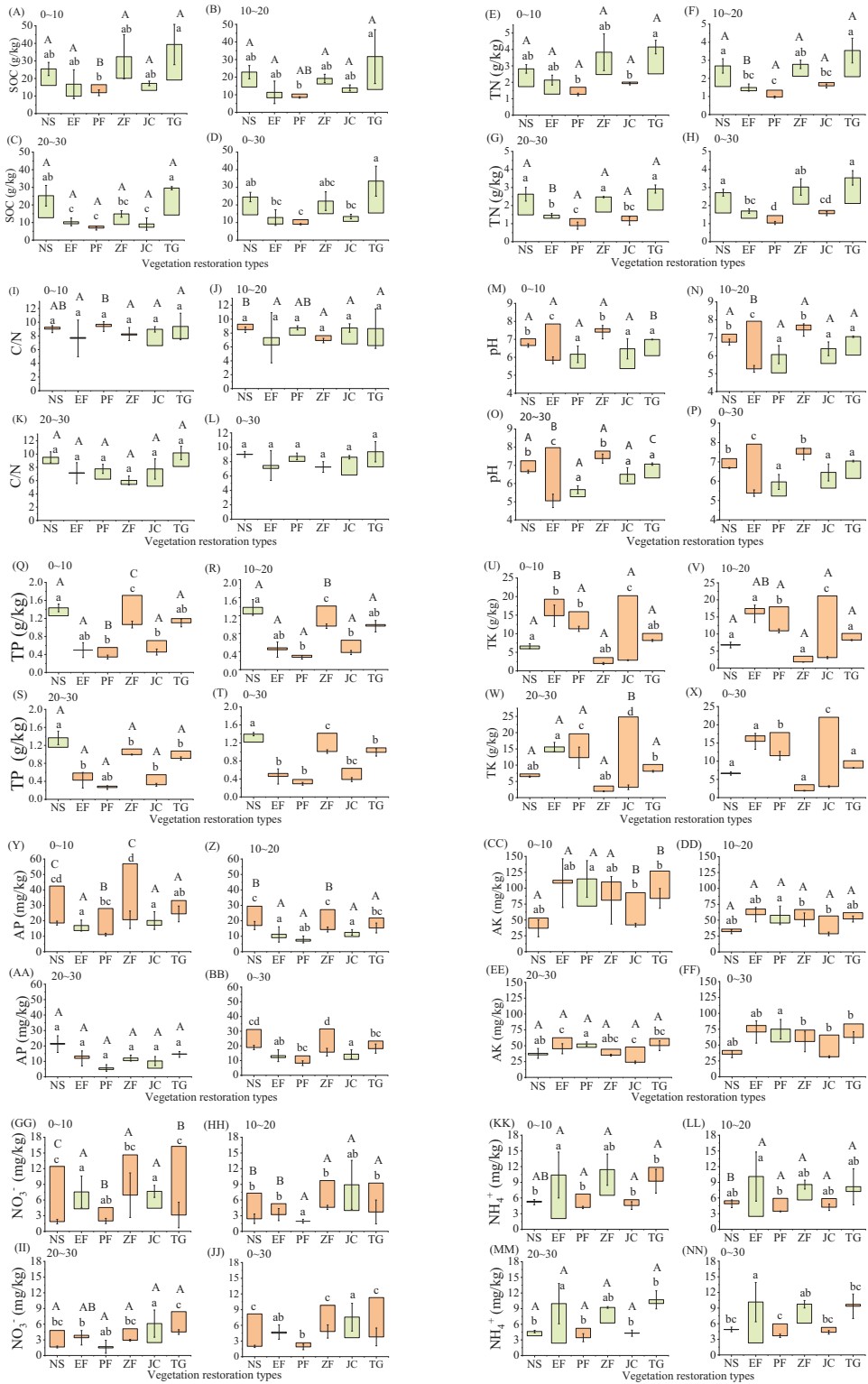

**Figure 2 Variation in soil physicochemical indicators under different types.** NS ($n = 3$), natural shrubland; EF ($n = 4$), *Eucalyptus robusta* economic forest; PF ($n = 3$), *Prunus salicina* economic forest; ZF ($n = 3$), *Zenia insignis* economic forest; JC ($n = 3$), *Juglans regia*-crop mixed forest; TG ($n = 3$), *Toona sinensis*- *Pennisetum purpureum* mixed forest. Data were obtained from three soil layers (0–10 cm, 10–20 cm and 20–30 cm) and their averages (0–30 cm). (A) soil organic carbon (SOC) at 0–10 cm,
**Table 5 Principal components analysis of soil properties.**

| Principal component | PC–1 | PC–2 | PC–3 | PC–4 | PC–5 |
|---|---|---|---|---|---|
| Eigenvalues | 3.59 | 2.63 | 1.76 | 1.52 | 1.07 |
| Variation (%) | 27.65 | 20.24 | 13.57 | 11.66 | 8.23 |
| Cumulative (%) | 27.65 | 47.89 | 61.46 | 73.12 | 81.35 |
| TN | 0.49 | −0.32 | **0.72** | 0.11 | −0.13 |
| SOC | **0.78** | −0.06 | 0.50 | 0.20 | 0.12 |
| C/N | 0.51 | −0.27 | 0.18 | 0.30 | 0.48 |
| TP | **0.77** | 0.33 | −0.08 | −0.37 | 0.10 |
| TK | **−0.72** | 0.41 | 0.11 | 0.32 | 0.13 |
| AP | **0.71** | 0.33 | −0.42 | 0.07 | 0.11 |
| AK | 0.22 | 0.56 | −0.10 | **0.51** | −0.17 |
| pH | 0.68 | −0.10 | −0.13 | **−0.53** | −0.08 |
| $NH_4^+$ | 0.32 | 0.35 | 0.31 | 0.46 | −0.48 |
| $NO_3^-$ | 0.49 | 0.21 | **−0.65** | 0.27 | −0.28 |
| clay | 0.04 | 0.62 | 0.00 | 0.14 | **0.58** |
| silt | −0.11 | 0.64 | 0.37 | −0.45 | −0.28 |
| sand | 0.07 | **−0.88** | −0.31 | 0.29 | −0.10 |

Note:
PC–1, PC–2, PC–3, PC–4 and PC–5 indicate the first to fifth principal component, respectively. Bold factors are considered highly weighted; TN indicates total nitrogen; SOC indicates soil organic carbon; C/N indicates the ratio of SOC to TN; TP indicates total phosphorus; TK indicates total potassium; AP indicates available phosphorus; AK indicates available potassium; $NH_4^+$ indicates ammonium-nitrogen and $NO_3^-$ indicates nitrate-nitrogen.

between −2.28 mg/kg and 7.84 mg/kg (Fig. 2NN). The $VNH_4^+$ values under *Eucalyptus robusta* was the largest and was significantly higher than that under restorations except *Zenia insignis* ($P < 0.05$).

## Evaluation of soil quality index

PCA based on all measured values, showed that the five main PCs with eigenvalues > 1 explained 81.35% of the total variation (Table 5). The major weighted indicators were

**Table 6 Pearson correlation coefficients of soil properties.**

| | TN | SOC | C/N | TP | TK | AP | AK | pH | $NH_4^+$ | $NO_3^-$ | clay | silt | sand |
|---|---|---|---|---|---|---|---|---|---|---|---|---|---|
| TN | 1.00 | | | | | | | | | | | | |
| SOC | 0.75** | 1.00 | | | | | | | | | | | |
| C/N | 0.31** | 0.61** | 1.00 | | | | | | | | | | |
| TP | 0.15 | 0.49** | 0.14 | 1.00 | | | | | | | | | |
| TK | −0.40** | −0.39** | −0.31** | −0.50** | 1.00 | | | | | | | | |
| AP | −0.07 | 0.35** | 0.29** | 0.67** | −0.34** | 1.00 | | | | | | | |
| AK | −0.05 | 0.18 | 0.07 | 0.07 | 0.19* | 0.36** | 1.00 | | | | | | |
| pH | 0.23 | 0.36** | 0.13 | 0.61** | -0.68** | 0.35** | −0.03 | 1.00 | | | | | |
| $NH_4^+$ | 0.34** | 0.35** | −0.03 | 0.18 | −0.02 | 0.17 | 0.34** | −0.08 | 1.00 | | | | |
| $NO_3^-$ | −0.22* | 0.10 | 0.06 | 0.34** | -0.28** | 0.61** | 0.40** | 0.27** | 0.26** | 1.00 | | | |
| clay | −0.15 | 0.04 | −0.03 | 0.26** | 0.27** | 0.20* | 0.24* | −0.08 | 0.15 | 0.04 | 1.00 | | |
| silt | −0.08 | −0.04 | −0.22* | 0.17 | 0.22* | −0.03 | 0.17 | 0.02 | 0.12 | −0.17 | −0.01 | 1.00 | |
| sand | 0.15 | 0.01 | 0.20* | −0.29** | −0.33** | −0.09 | −0.28** | 0.03 | −0.18 | 0.12 | −0.56** | −0.82** | 1.00 |

Notes:
  * Correlation is significant at $P < 0.05$ level.
  ** Correlation is significant at $P < 0.01$ level.

**Table 7 Normalization equation of scoring curves.**

| Parameter | TN | SOC | TP | TK | AK | pH | $NO_3^-$ | clay | sand |
|---|---|---|---|---|---|---|---|---|---|
| Average (x0) | 1.29 | 16.25 | 0.82 | 10.09 | 59.96 | 6.45 | 5.43 | 19.43 | 18.01 |
| Slope (b) | −2.5 | −2.5 | −2.5 | −2.5 | −2.5 | −2.5 | −2.5 | −2.5 | 2.5 |
| Weighting value | 0.16 | 0.21 | 0.14 | 0.06 | 0.13 | 0.08 | 0.05 | 0.09 | 0.08 |

SOC, TP, TK and AP in PC1, sand was the only suitable indicator in PC2, TN and $NO_3^-$ were selected for PC3, AK and pH were selected for PC4, and clay was selected for PC5. Significant correlation > 0.6 ($P < 0.05$) was observed between AP and TP in PC1, and AP was removed (Table 6). Thus, the precise MDS contained nine indicators: TN, SOC, TP, TK, AK, pH, $NO_3^-$, clay and sand. The weighting values were analyzed by PCA based on the MDS measured values (Table 7). Then, SQI was calculated by Eq (4):

$$SQI = (0.16 \times TN) + (0.21 \times SOC) + (0.14 \times TP) + (0.06 \times TK) + (0.13 \times AK)$$
$$+ (0.08 \times pH) + (0.05 \times NO_3^-) + (0.09 \times clay) + (0.08 \times sand) \tag{4}$$

Natural shrubland, *Toona sinensis-Pennisetum purpureum* and *Zenia insignis* increased SQI significantly ($P < 0.01$) comparing to the cropland at 0–30 cm soil layer, with increasing value 0.17, 0.17 and 0.10, respectively. Furthermore, natural shrubland had significant higher SQI at all three soil layers, while *Prunus salicina* only improve SQI at 10–20 cm soil layer ($P < 0.05$) (Table 8). Variation in SQI (VSQI) of natural shrubland in total soil depth (0–30 cm) was highest and significantly higher than that of *Eucalyptus robusta*, *Prunus salicina* and *Juglans regia*-crop, while *Juglans regia*-crop with lowest VSQI at every soil layer significantly lower than that of natural shrubland and

**Table 8 Paired *t* test values for effect on soil quality index (SQI) between vegetation restoration type and cropland.**

| Set | 0-30 cm | 0-10 cm | 10-20 cm | 20-30 cm |
|---|---|---|---|---|
| NS–CK1 | ++ | + | ++ | + |
| EF–CK2 | | | | |
| PF–CK3 | | | + | |
| ZF–CK4 | ++ | | | |
| JC–CK5 | | | | |
| TG–CK6 | ++ | | + | |

**Note:**
++: a significant increase at $P < 0.01$ level in double-tailed $t$ test. +: a significant increase at $P < 0.05$ level in double-tailed $t$ test, empty shows no significant change.

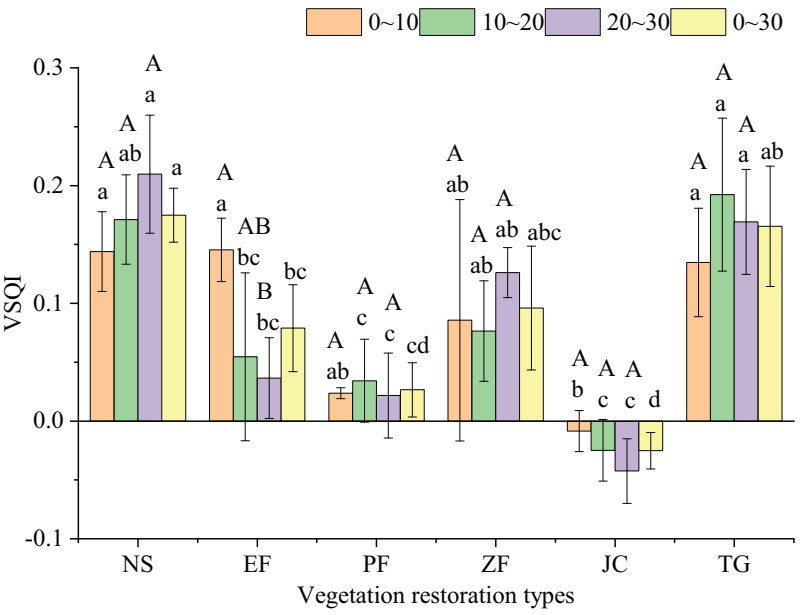

**Figure 3 Variation in soil quality index under different restoration types.** VSQI, variation in soil quality index. Different lowercase letters indicate significant difference among different vegetation restoration types at the same depth (one–way ANOVA, $P < 0.05$) and different uppercase letters indicate significant difference under different soil depths at the same type (one–way ANOVA, $P < 0.05$).

*Toona sinensis-Pennisetum purpureum* (Fig. 3) ($P < 0.05$). Significant difference of VSQI among soil depth only found in *Eucalyptus robusta*, which decreased with increasing soil depth ($P < 0.05$).

In the BRT model, vegetation restoration type, restoration time (year), surface vegetation coverage, soil depth, latitude, longitude, elevation and slope of the study site were considered for analyzing the variation in VSQI (Fig. 4). Vegetation restoration type explained the largest proportion of VSQI variation (73.49%), followed by restoration time (10.30%), soil depth (7.74%), and latitude (5.31%), with the other four factors having little effect on VSQI.

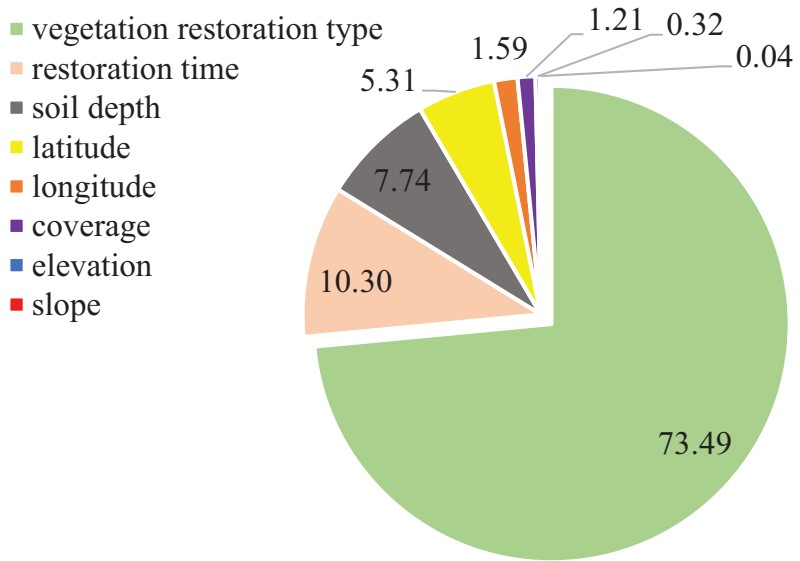

**Figure 4 Effects on the variation soil quality index by the boosting regression tree model.** Results obtained from boosting regression tree model (BRT) showed the integrative effects of vegetation restoration type, soil depth, restoration time, latitude, longitude, elevation, coverage and slope on the variance soil quality index.

## DISCUSSION

### The effects of vegetation restoration on soil quality compared to cropland

The effects of vegetation restoration on soil quality in degraded lands remain controversial. Some studies reported that vegetation restoration could improve soil physicochemical properties in karst cropland, such as increasing soil nutrients like SOC, N concentration (*Chen et al., 2019*; *Liu et al., 2019*; *Zhang et al., 2019a*); others supported that most of the restoration projects were failed or get limited success (*Asmelash, Bekele & Birhane, 2016*), especially in the fragile karst ecosystem southwest China, where an ecological 'tipping point' may have been passed, beyond which soil properties are unrecoverable in manageable timescales (*Guo et al., 2019*). In our study, none of the vegetation restoration types decreased soil quality significantly, with VSQI between −0.03 and 0.17, SQI values in natural shrubland, *Toona sinensis-Pennisetum purpureum* and *Zenia insignis* in total soil depth (0–30 cm) were significantly higher than that in the corresponding controls (Table 8). This may be due to that *Toona sinensis-Pennisetum purpureum* and natural shrubland all significantly increased SOC and TN compared to cropland (Table 3), and *Zenia insignis* also achieved a non-negligible increase in SOC and TN (Fig. 2), which were the dominant indicators to developing SQI in this study. Our findings are in accordance with the previous results that levels of SOC and TN in natural restoration vegetation (*Chen et al., 2012*; *Wang et al., 2018*) and combination of plantation trees and forage grasses (*Xiao et al., 2017a*; *Hu et al., 2019*) were significantly higher than the corresponding values in cropland due to greater litter input in these restoration types. On the other hand, SOC in the cropland decomposed rapidly as a result of land cultivation

(*Zhu et al., 2014*), which leads to more severe reduction in subtropical karst area (*Jiang et al., 2006*). No significant increase of SOC were found in the remaining vegetation restorations, probably because tree species in these restorations were fast-growing and not conducive to the accumulation of SOC. In addition, *Eucalyptus robusta* and *Juglans regia*-crop, significantly increase sand content that weaken soil quality (Table 2), similar to the previous studies showed that sand content in woodlands was higher than that in adjacent cropland (*Qin et al., 2017*; *Xiao et al., 2017b*). Moreover, the restoration time of *Juglans regia*-crop (1 year) was too short to make significant accumulation of soil organic matter. Thus, no significant improve of SQI were observed in these 2 restorations compared to cropland.

VSQI in *Eucalyptus robusta* decreased with increasing soil depth ($P < 0.05$), which consistent with previous studies that soil quality was higher in surface soil layer than that in the deeper layers, due to the litter accumulating on the surface and then transformed into nutrients with microbial activity (*Mukhopadhyay et al., 2016*; *Zhang et al., 2019a*). However, we found that the improvement of soil quality in subsoil was better than that of surface soil in natural shrubland, *Toona sinensis-Pennisetum purpureum* and *Prunus salicina* (Table 8), mainly due to the variation in $NO_3^-$ and AK increased with increasing soil depth in these restoration types (Figs. 2CC–2JJ), as a result of the surface soil of cropland being nutrient-rich from application of inorganic fertilizers. On the other hand, the subsoil of unplowed soil has a considerable capacity to adsorb P and K, resulting in increasing concentrations of these nutrients in subsoil (*Zhang et al., 2013*; *Roy & Bickerton, 2014*), and resulting in lower reductions or even increases in nutrient concentrations in subsoil of restorations when comparing to cropland. Over all, this study demonstrated that natural shrubland, *Toona sinensis-Pennisetum purpureum* and *Zenia insignis* could improve soil quality in degraded karst cropland, while *Prunus salicina* only make efficiency in subsoil.

## Analysis of the effects of different vegetation restoration types on soil quality

Previous studies confirmed that the ability of different vegetation restoration types to improve soil quality are different (*Yu et al., 2018*; *Dang et al., 2020*; *Liu et al., 2020*). In this study, the effect on soil quality (0–30 cm) of natural shrubland was significantly higher than that of *Eucalyptus robusta, Prunus salicina* and *Juglans regia*-crop (Fig. 3), mainly due to the significant higher of variation of SOC, TN, TP and TK in natural shrubland (Fig 2D, 2H, 2T and 2X). The benefits of vegetation restoration evaluated by the value of the difference between vegetation restoration and the corresponding farmland has similar results to those of direct value responses, recent relevant studies, such as *Yang et al. (2017)* reported that SOC and TN in natural reserve were higher than that in economic forests (*Zenia insignis, Toona sinensis* and orchard); *Tang et al. (2015)* found that natural successional plant communities had higher soil fertility parameters, such as SOC, AN, AP and AK, as compared with *Pinus* plantations; *Pang et al. (2019)* proved that forest natural regeneration was more effective on SOC sequestration than *Pinus* and *Eucalyptus* plantation. These findings can be explained by the fact that less human

disturbance and higher soil nutrients input supplied by dead wood and leaf litter in natural recovery community (*Li et al., 2018b*; *Shen et al., 2020*). In contrast, *Eucalyptus robusta*, *Prunus salicina* and *Juglans regia*-crop have limited influence on soil improvement mainly resulted by their fast growing and high output that would accelerate absorption of soil nutrients (*Laclau et al., 2005*; *Zhang et al., 2015*; *Macdonald et al., 2019*). Furthermore, plantation of *Eucalyptus robusta* greatly reduced soil pH, this acidifying effect was also reported by lots of studies (*Rhoades & Binkley, 1996*; *Soumare et al., 2016*; *Zhang et al., 2019b*), which may subsequently cause soil nutrient deficiency (*Banfield et al., 2018*), supported by our result that pH was positively correlated with most nutrient contents (Table 6).

Variation in soil quality was affected by many factors, but only eight main factors were considered in this study. The result of boosting regression tree model showed that restoration type had the greatest contribution (73.49%) to the final VSQI, which was in accordance with previous studies that vegetation type is the key factor affecting soil quality (*Fan et al., 2019*; *Yan et al., 2020*). Therefore, the selection of the most appropriate restoration type for vegetation restoration in karst areas is crucial in terms of effectively improving soil quality. Followed by restoration time (10.30%), soil depth (7.74%), and latitude (5.31%), together contributing to 96.84% of VSQI variation. Similarly, *Zhang et al. (2019a)* found that restoration type and soil depth were the two most important factors contributing to SQI of vegetation restoration in degraded karst areas. In addition, restoration time played a non-negligible role in VSQI, due mainly to ecosystem carbon stock sequestration achieved over time as a result of vegetation restoration (*Zhang et al., 2018a*), since soil organic matter/carbon is the key factor to determining soil quality (*Bunemann et al., 2018*). In conclusion, vegetation restoration type is the dominant factor that affects soil quality and the longer restoration time achieves the better restoration effects. Thus, natural shrubland has better capacity to recover soil quality than *Eucalyptus robusta*, *Prunus salicina* and *Juglans regia*-crop in karst regions.

## CONCLUSIONS

Effect of vegetation restoration on soil quality was evaluated by the soil quality index, and the benefits of different vegetation restoration types were compared by the value of the difference between vegetation restoration and the corresponding cropland (VSQI). In summary, restoration type accounted for the most variation (73.49%) of the VSQI, and then restoration time (10.30%), soil depth (7.74%), it is of great significance to select suitable vegetation types and last for a long time for restoration to improve soil quality. Natural shrubland had better capacity in improving soil quality in degraded karst cropland than *Eucalyptus robusta*, *Prunus salicina* and *Juglans regia*-crop, mainly due to its higher SOC and TN content. Among them, natural shrubland and *Toona sinensis-Pennisetum purpureum* were the most effective restoration type for degraded cropland. Further research should examine the impacts of soil biological properties in soil quality index and take the structure and composition of each vegetation community into consideration.

### Funding
This work was supported by the National Key Research and Development Program (2017YFC0506505). The funders had no role in study design, data collection and analysis, decision to publish, or preparation of the manuscript.

### Grant Disclosures
The following grant information was disclosed by the authors:
National Key Research and Development Program (2017YFC0506505).

### Competing Interests
The authors declare that they have no competing interests.

### Author Contributions
- Huiling Guan conceived and designed the experiments, performed the experiments, analyzed the data, prepared figures and/or tables, authored or reviewed drafts of the paper, and approved the final draft.
- Jiangwen Fan conceived and designed the experiments, authored or reviewed drafts of the paper, and approved the final draft.

### Data Availability
Raw data are available in a Supplemental File.

### Supplemental Information
Supplemental information for this article can be found online at http://dx.doi.org/10.7717/peerj.9456#supplemental-information.

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
