# Peer review of "Effects of vegetation restoration on soil quality in fragile karst ecosystems of southwest China"

_PeerJ, doi:10.7717/peerj.9456_

## Round 0.1 · original submission · Major Revisions

Dear Jiang wen Fan,

My name is Leonardo Montagnani, and I have taken charge as editor of your manuscript, replacing the originally assigned Editor. Sorry for the delay which has occurred.

I read the two reviews of your manuscript, indicating that major revision is required. In particular, I also agree with one reviewer that the experimental set-up is problematic. Since it is not possible now to make new soil analyses along the restoration process, it is essential that you only draw conclusions that you can obtain with the current set-up.

Reviewer 1 ·

Basic reporting

Understanding the effect of restoration activities on soil quality is a relevant issue, especially in karst ecosystems with erosion prone soils. However, the analysis presented cannot support the inferences made in the study.

Experimental design

My major concern is the experimental design. Need a better description of the study sites, including the size of each restoration site and the separation within the each plot of the restoration treatment to ensure independence of the each sample. The study tries to compare different restoration techniques, but to make this comparison; restoration treatments should have replication.

Validity of the findings

One possible analysis that could be made is the effect of a pre-restored (control sites) and restored sites. Even this approach will need the standardization of the variability of the restoration techniques. Another possible analysis with the current data will be to compare the treatment of economic forests with natural restoration techniques and the control. The analysis of the Soil Quality Index still could be use the PCA, but using the treatments previously suggested.

I am aware that having replication in restoration projects is always a limitation. Therefore, it is a good practice to avoid the use of parametric test, One-way analysis of variance, if the initial project did not consider the treatments during the restoration activity.

Reviewer 2 ·

Basic reporting

no comment

Experimental design

no comment

Validity of the findings

no comment

Additional comments

1. The workload of the article is relatively large, but thinking and logicality of the article have some deficiencies. The article needs major revision.
2. Why do you choose the cropland as control and how to consider the effect of vegetation.
3. The research of this article is of great significance, but the conclusion was just the collection of results. A high quality of conclusion should be summarized and abstracted form these results. The authors need to refine the research results to enhance the significance of the research.
4. The latest research results should be selected for literature citation as far as possible.
5. What's the purpose of the first paragraph of the introduction? It seems not closely related to the content of this article.
6. Line 70, the hypothesis is not consistent with the objectives of this research.
7. Line 79-80, the reasons of choosing this index were not explained in the introduction of this paper. Suggest to cite relevant references to explain.
8. The spatial distribution of sites is far away, and how to exclude the influences of uneven elements such as climate.
9. Why did this research studied three soil layers and it need to be explained and discussed deeper.
10. The discussion of this article should be discussed deeper.

---

## Round 0.2 · Minor Revisions

Dear Dr. Guan,

Your paper is improved, but I believe that is not ready for publication.
There are several open issues.

Please improve the language, with the help of a professional editing service. Check the coherence in the writing, tables and similar. The scientific binomial name of a plant requires the capital letter in the first name.

Possibly, more importantly, check when you mention restoration, it is really the result of a process that lasted some years (how many?) or you are describing the soil types belonging to different land use.

Increase the font size in Figure 4.

I did not mention all the details that have to be fixed, so please check carefully.

I expect your revised text version.

Sincerely,

Leonardo Montagnani

Reviewer 2 ·

Basic reporting

no comment

Experimental design

no comment

Validity of the findings

no comment

Additional comments

I think the author has adopted the opinions of the reviewers and carefully revised the manuscript. The manuscript has been significantly improved, and the quality is now suitable for publication in terms of ecology and environment.

---

## Round 0.3 · Minor Revisions

Dear Jiang wen Fan,

I considered again your manuscript. I found that it is improved in several details but unfortunately not in one key aspect. I believe that when you are considering the effect of a restoration activity you cannot separate the time from the restoration type. I believe that to properly assess the effect of the proposed land-use change we always have to consider the starting point, the years between the start and the final point.

For instance, if you have a site starting from degraded land becoming a forest preserve, you cannot use an agricultural field nearby as a starting point, but another degraded land. Then, you can compute the annual variation in soil properties characterizing that specific variation.
In my perspective it is essential that you do not you perform a statistic based on the wrong assumption. It is better that you simply report a table clarifying what is the starting point in terms of land use, the years passed and the final point. Presenting the nearby agricultural field is not necessarily the correct starting point, only in some cases.

I add a couple of other indications:

To complete the manuscript, please use a personal computer with standard fonts, some were not correctly visualized in the last version;

Please use the scientific binomial nomenclature for plant names in all the occurrences, including tables. Try to uniform the categories (for example, not plum…Juglans regia).

I expect you to send the revised version of your manuscript with the requested modifications. If you disagree, you must provide convincing arguments.

Sincerely,

Leonardo Montagnani

---

## Round 0.4 · accepted · Accept

Dear Jiang Wen Fan,

I am pleased to inform you that I consider your paper acceptable now.

Sincerely,

Leonardo Montagnani